# Efficient Video-Based Deep Learning for Ultrasound Guided Needle Insertion

**Jonathan Rubin**[*]                               JONATHAN.RUBIN@PHILIPS.COM

**Alvin Chen**[*]                                   ALVIN.CHEN@PHILIPS.COM

**Anumod Odungattu Thodiyil**                       ANUMOD.T@PHILIPS.COM

**Raghavendra Srinivasa Naidu**                     RAGHAVENDRA.SRI@PHILIPS.COM

**Ramon Erkamp**                                    RAMON.ERKAMP@PHILIPS.COM

**Jonathan Fincke**                                 JONATHAN.FINCKE@PHILIPS.COM

**Balasundar Raju**                                 BALASUNDAR.RAJU@PHILIPS.COM

*Philips Research North America*

*Precision Diagnosis and Image Guided Therapy Department*

**Editors:** Under Review for MIDL 2021

## Abstract

We investigate video-based deep learning approaches for detecting needle insertions in ultrasound videos. We introduce two efficient and conceptually simple extensions to convert standard 2D object detectors into video object detectors that encode spatiotemporal information from a history of frames. We demonstrate that learned temporal encoding improves needle detection over spatial encoding alone, particularly for challenging cases. Finally, we show that the proposed methods allow real-time inference on low-cost computing hardware.

**Keywords:** Ultrasound, video object detection, guided needle visualization

## 1. Introduction

Ultrasound is frequently used as a non-invasive and real-time imaging method to guide needle insertion procedures. Nevertheless, needle visualization under ultrasound can often be difficult due to off-axis specular reflections, speckle noise, tissue structures with needle-like appearances, and poor needle visibility. Visualization is particularly challenging when the needle is inserted very steeply, just entering the imaging field, or partially out-of-plane (Fig. 1). Earlier studies proposed a range of image enhancement and machine learning approaches operating on individual ultrasound frames (for a survey, see Beigi et al., 2021). However, recognizing the needle from single frames can be difficult due to ambiguity relative to surrounding tissues. Several studies have addressed this by including engineered temporal features such as image differencing, optical flow, or time-phase signatures (Beigi et al., 2021).

We build on the prior body of work by investigating video-based deep learning approaches that extract combined spatial and temporal features for needle localization in an end-to-end manner. Specifically, we introduce two efficient and conceptually simple extensions to convert standard 2D object detectors into video object detectors operating on a history of frames. We compare our approaches to a 2D baseline method that makes independent predictions per frame, and we demonstrate that spatiotemporal feature encoding improves needle detection over spatial encoding alone, particularly for challenging cases such as short and steep insertions. Given the need to run in real-time on computationally restricted environments, emphasis is placed on speed and low computational complexity.

---

[*] Contributed equally

Figure 1: Needle insertions highlighting the difficultly of localizing from a single frame.

## 2. Methods

**2D Detector** Baseline approach that ignores temporal information and applies standard 2D object detection independently to each video frame (Fig. 2(a), left).

**2.5D Detector** Early fusion approach that feeds both image and motion information to a network (Fig. 2(a), middle). For the motion stream, we introduce a *k compounded difference* approach which combines the difference between the current and $k$ previous frames into a 2D motion map, $I_{motion}^t = \sum_{i=1}^{k} f * (I_t - I_{t-i})^2$, where $f$ is a denoising or smoothing operation. We use Gaussian smoothing and set $k=6$. We concatenate $I_{motion}^t$ with the image frame $I_t$ along the channel dimension and feed to a 2D detector network.

**3D Detector** End-to-end spatiotemporal learning approach that prepends a 3-dimensional spatiotemporal convolutional block to the head of an existing object detector to allow temporal information to inform bounding box predictions (Fig. 2(a), right). 3D convolution (i.e. 2 spatial, 1 temporal) is applied repeatably until a single temporal dimension remains. We chose input video clips consisting of $k=6$ frames. Thus we applied $(k$-1)@2x3x3 3D kernels (depth x height x width) with 0x1x1 padding. After $k$-1 convolutions, we arrive at a single feature map encoding learned temporal information, which is fed to the 2D detector. Predictions are made for the current frame of the input video clip, $I_t$.

For each approach, we further evaluate the following. **Augmentation:** we apply image, spatial and temporal augmentations including gain, contrast, and noise adjustments, as well as time reversal, compression and dilation. **Focal loss:** we use focal loss with $\gamma=1.5$ to focus on more difficult examples during training. **Geometric prior:** we make use of known pose constraints (needle orientation and intercept) and penalize predictions far away from known pose distributions derived from training labels.

We require real-time detection, including on low-cost ultrasound CPU hardware. We found that Faster R-CNN (Ren et al., 2015) and more recent self-attention based approaches (Carion et al., 2020) were unable to meet these requirements. Instead, we used the Yolov3-tiny architecture (Redmon and Farhadi, 2018) as a backbone network and further restricted network sizes by reducing the number of channels per layer by a factor of eight.

## 3. Results

All models were trained on a data set of 30,072 mini video clips. 5,133 additional clips were used for evaluation. In both sets, 60% of frames were labeled with the needle present, and the remaining 40% were negative. Clips were acquired on ex-vivo (porcine/bovine/chicken)

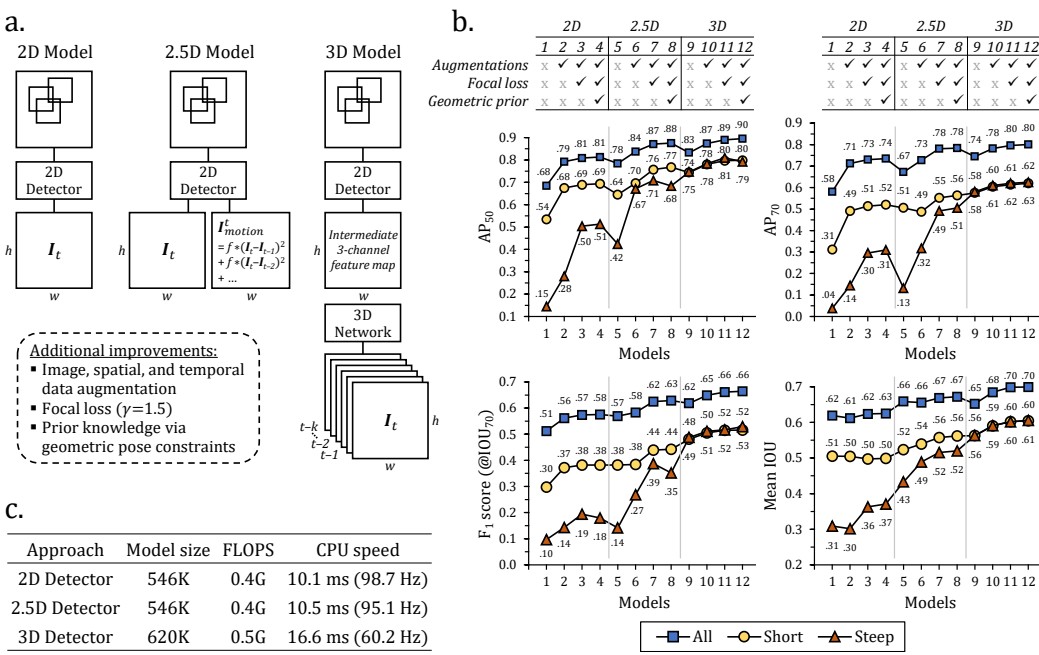

Figure 2: (a) Video object detection networks for needle localization. (b) $AP_{50}$, $AP_{70}$, $F_1$ and IoU for each approach. (c) Computational complexity for each approach.

and cadaver tissues using 12 and 18 MHz transducers and 20 to 22 gauge needles. We defined challenging cases as either short ($\leq$5 mm distance) or steep ($\geq$60° angle) insertions.

Fig. 2(b) shows $AP_{50}$, $AP_{70}$, $F_1$ and IoU for the three approaches in Fig. 2(a), together with the proposed strategies for data augmentation, focal loss, and geometric pose constraints. The 2.5D and 3D video detectors, as well as added data and training strategies, were found to improve significantly on the baseline 2D detector. The impact of these methods were particularly notable for challenging cases (short and steep insertions).

Fig. 2(c) presents the computational performance (Intel Core i5, 2.9 GHz speed) for each of the three architectures. Real-time inference was achieved in all cases, with a small increase in processing time for the 3D network. The results indicate that the proposed methods can be implemented on low-cost ultrasound processing hardware to facilitate clinical translation.

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
