# OpenReview forum: "Efficient Video-Based Deep Learning for Ultrasound Guided Needle Insertion"
_MIDL.io/2021/Conference/Short — MIDL 2021 Poster_

### Official Review · Reviewer_Hrzg · 2021-04-27

**Confidence:** 4
**Final Rating:** 3

**Summary:**

This submission presents the study of three deep learning solutions for needle detection in 2D ultrasound videos. Particularly, the paper analyzes a standard 2D object detection pipeline in comparison with two improved strategies that exploit (i) the visual information from the current frame and the motion information coming from previous frames, (ii) the information of multiple stacked frames processed via 3D convolutions.  All the solutions are based on the Tiny Yolo object detector architecture. The results show that the 3D-based solution outperforms the other strategies under different scenarios and training configurations. Real-time speed is achieved by all the detectors.

**Strengths:**

- **Problem** The problem tackled is relevant and very challenging. Figure 1 shows that it is very hard to distinguish needles from other objects appearing in the ultrasound frames.
- **Application** The paper applies several deep learning video object detection strategies to a new problem.
- **Technical contribution** The baseline Tiny Yolo is improved with two different strategies that carry some technical contribution from the ultrasound temporal feature learning perspective.
- **Analysis** The paper analyses three strategies for video object detection under different perspectives and learning configurations. Overall, it points out interesting insights on the application of these models to the problem of needle localization.
- **Dataset** The dataset used for the validation and study of the solutions is impressive.
- **Presentation** The paper is well written and the concepts are clearly expressed. A good amount of detail is given even the limited text space.

**Weaknesses:**

- **Motivation** The paper does not justify the employment of video object detection solutions for the problem of interest. Why do the authors prefer such a strategy for needle localization in contrast to e.g. a detection+tracking pipeline? Trackers have been extensively used in many ultrasound applications [1,2,3], demonstrating more efficacy in the exploitation of temporal information.

[1] Combining multiple dynamic models and deep learning architectures for tracking the left ventricle endocardium in ultrasound data, Carneiro and Nascimento, TPAMI 2013

[2] Siamese Networks with Location Prior for Landmark Tracking in Liver Ultrasound Sequences, Gomariz et al., ISBI 2019

[3] Siam-U-Net: encoder-decoder siamese network for knee cartilage tracking in ultrasound images, Dunnhofer et al., MedIA 2020


**Deanonymize Review:**

no

**Detailed Comments:**

It would be nice to have in the paper some qualitative examples of the performance of the three solutions. Additionally, it would interesting to know the average length in frames of the clips.

**Justification Of The Rating:**

This paper tackles a challenging problem and studies the application of well-known solutions (Tiny Yolo) with incremental technical improvements. Even though the paper does not justify the employment of video object detectors rather than other object localization strategies, I believe the extended analysis can give interesting insights on the performance of such models, making it a valuable point of discussion. Hence, I recommend its acceptance.

**Paper Type:**

validation/application paper

**Special Issue:**

no

---

### Official Review · Reviewer_MMF9 · 2021-04-28

**Confidence:** 3
**Final Rating:** 2

**Summary:**

This paper proposes to use real-time 2.5D or 3D methods for detecting needle insertions in ultrasound videos. Compared to 2D methods operating on individual ultrasound frames, the proposed 2.5D or 3D methods encode temporal information to improve detection on a 2D frame, utilizing simple feature fusion strategies (Gaussian smoothing on frame differences for the 2.5D method, and 3D conv for the 3D method) on a previous frame sequence and the current 2D frame. The experimental results shows the efficacy of the proposed 2.5D and 3D methods.

**Strengths:**

1. The paper is clearly written and well organized.
2. The proposed 2.5D/3D methods and the additional operations (augmentation, focal loss, and geometric prior) are easy to use.
3. The experimental results shows the effectiveness of the proposed components and the 3D method with all additional operations achieve the best performance.

**Weaknesses:**

1. The current task is essentially a 2D detection problem. The evaluation metrics do not evaluate performance on video level. Therefore, the reviewer is not sure whether this can count as validation/application contribution.
2. The proposed 2.5D/3D methods and additional operations seem to be trivial application of existing works.
3. This paper did not consider video object detection methods, but only compared to a simple 2D detection baseline.

**Deanonymize Review:**

no

**Justification Of The Rating:**

This paper proposes effective 2.5D and 3D pipeline for the task of detecting needle insertions in ultrasound videos. However, as discussed in weaknesses above, the novelty of this work seems to be limited.

**Paper Type:**

methodological development

**Special Issue:**

no

---

### Meta-Review · Area_Chair_8nww · 2021-05-10

**Recommendation:** Accept (Poster)
**Confidence:** 5

**Metareview:**

The reviewers point out the limited methodological contribution and suboptimal connection to the video setting. At the same time, they applaud the presentation and general usefulness of the approach. Given that the paper could provide valuable insights for other practitioners with similarly challenging data and strict requirements on run time, it can be accepted.

---

### Decision · Program_Chairs · 2021-05-11

Accept (Poster)